# Pitfalls in Evaluating Language Model Forecasters

**Daniel Paleka**[*1]   **Shashwat Goel**[*2,3]   **Jonas Geiping**[2,3]   **Florian Tramèr**[1]

[1]ETH Zurich   [2]ELLIS Institute Tübingen   [3]MPI Tübingen

## Abstract

Large language models (LLMs) have recently been applied to forecasting tasks, with some works claiming these systems match or exceed human performance. In this paper, we argue that, as a community, we should be careful about such conclusions as evaluating LLM forecasters presents unique challenges. We identify two broad categories of issues: (1) difficulty in trusting evaluation results due to many forms of temporal leakage, and (2) difficulty in extrapolating from evaluation performance to real-world forecasting. Through systematic analysis and concrete examples from prior work, we demonstrate how evaluation flaws can raise concerns about current and future performance claims. We argue that more rigorous evaluation methodologies are needed to confidently assess the forecasting abilities of LLMs.

## 1 Introduction

Forecasting, the task of assigning probabilities to future events, represents a critical capability for decision-making across various domains. Several recent studies have explored the potential of LLMs as forecasting systems, sometimes even suggesting that LLMs can already rival human performance (Halawi et al., 2024; Phan et al., 2024; Schoenegger et al., 2024).

However, this paper identifies significant concerns in the trustworthiness of reported results for both existing and future LLM forecasting systems. We expand on several issues that are partially known in the forecasting community (e.g., (Bosse et al., 2024; Sempere and Lawsen, 2021; Arnott et al., 2018)), but have not been comprehensively analyzed for LLM forecasters. Through concrete examples, we illustrate subtle challenges in forecasting evaluation, and discuss how these issues may have led to overly optimistic assessments of LLM forecasting abilities in prior work. The challenges we identify fall into two broad categories:

1. *Trusting evaluation results*: Various forms of data leakage make it difficult to ensure that models truly predict future events rather than using information after the forecast date.

2. *Extrapolating from evaluation results to real-world performance*: Good performance on forecasting benchmarks may not necessarily correspond to good forecasting abilities.

We provide an overview of these challenges below, and analyze them in Sections 2-3.

**Challenge 1: Establishing trustworthy evaluation results.**   The gold standard for evaluating a forecaster involves running it on unresolved questions, waiting until the questions resolve, and then scoring the predictions. However, this approach is impractical for rapid model evaluation. Thus, researchers typically resort to *backtesting* or *retrodiction* (Zou et al., 2022), where the forecasting system is given knowledge as of some past time $T$ and asked to forecast events between time $T$ and the present. Although appealing in principle, backtesting introduces several issues:

- *Logical leakage:* The very nature of backtesting can logically constrain possible answers. Consider a time traveler analogy: If someone from 2035 asks you to predict if we will find alien life before 2040, you can deduce that the answer must be "yes"–otherwise, the time traveler would not yet have definitive evidence to grade your prediction. We show that a significant percentage of questions in prior forecasting benchmarks permit similar logical deductions of the correct answer.

---

*Equal contribution.

- *Unreliable date-restricted retrieval*: Many forecasting systems incorporate retrieval components (e.g., search engines) that are restricted to data available at time $T$. Yet, date metadata on documents is often inaccurate, allowing future data to leak into the system. More subtly, the retrieval model itself has typically been trained on future data, causing leakage through learned associations. For example, searching for "January 6th" with a date restriction to 2020 returns documents with an abnormally strong association to U.S. politics (by 2020 standards).

- *Over-reliance on model cutoff dates*: Researchers often assume that models have no knowledge beyond their reported training cutoff dates. However, cutoffs are more of a guideline than a guarantee, and evidence suggests that models possess knowledge of some events beyond these dates.

**Challenge 2: Extrapolating from benchmark performance.** Even with a sound evaluation, translating results into real-world forecasting ability faces additional issues:

- *Piggybacking on human forecasts*: Many forecasting datasets originate from human prediction platforms. Thus, human forecasts are likely available to the LLM (in its training data or through its retrieval system). Claims that LLMs "match human performance" may then be circular: the models might simply be copying human forecasts rather than demonstrating independent ability.

- *Gaming benchmarks through betting*: Unlike many AI tasks, forecasting benchmarks can reward strategic gambling over accurate uncertainty estimation. For example, consider forecasting U.S. politics in 2025, from a vantage point in 2023 with a 50/50 prior on the presidential election. The benchmark-optimal strategy might be to commit to one outcome and condition all predictions on that guess. With 50% chance, this yields excellent performance; with 50% chance, it fails completely. This has a much higher chance of topping the benchmark than a calibrated strategy.

- *Skewed data distribution*: Questions on forecasting platforms often focus on topics that competitive forecasters find interesting, creating potential distribution biases. When curating benchmarks for backtesting, these biases can be exacerbated by constraints on which questions can be resolved within the evaluation timeframe. Although data biases exist in many ML benchmarks (e.g., ImageNet's focus on dog breeds still produces transferable visual features), there is little evidence that performance on current forecasting benchmarks yields generalizable forecasting capabilities.

**Looking ahead: Challenges in optimizing better forecasters.** Current LLM forecasters primarily use off-the-shelf models. As the field advances, a natural next step will be applying optimization pressure specifically to improve forecasting performance. However, the temporal nature of the data makes this optimization challenging. Naively training on question-answer pairs over a time period creates temporal leakage, as early samples in training can leak information relevant to later ones (e.g., "Who will win the election?" followed by "Who will win the primaries?"). Temporally sorting training samples still fails to properly simulate the task of predicting events further in the future.

By developing these arguments, we argue that the monitoring of forecasting capabilities is appreciably harder than the evaluation of knowledge about the past or present, as it has unique issues on top of all the existing issues with machine learning evaluation (Leech et al., 2024). Forecasting future events is a fascinating and challenging task for LLMs, with wide-ranging implications, and we hope to convince the community to devote more attention to evaluating it carefully.

## 2 CHALLENGE 1: ESTABLISHING TRUSTWORTHY EVALUATION RESULTS

The ideal way to evaluate a forecasting system is to pose questions about future events, collect predictions, and score them as the events resolve. However, such evaluations can take months or years. In practice, researchers use *backtests*, which constrain a system's knowledge to information available up to some past time $T$, and then pose questions that resolved between $T$ and the present (Tashman, 2000). This allows for immediate feedback, enabling rapid iteration. However, the assumption that the system lacks information after the chosen time $T$ is often violated in subtle ways.

### 2.1 ISSUE 1: LOGICAL LEAKAGE OF OUTCOMES IN BACKTESTING

When backtesting forecasters, we select (or generate) forecasting questions at some past time $T$ *with knowledge of what the future holds.* Natural strategies for selecting forecasting questions can thus implicitly leak information about the future, if the forecaster knows that it is being backtested.

**Knowledge of the backtesting date can leak outcomes.** It may seem reasonable to collect questions from human forecasting platforms, with the restriction that (1) the question was formulated before time $T$; and (2) the question has resolved in the present. Yet, the latter condition can implicitly leak the answer: Suppose in 2021 we asked "Will Queen Elizabeth live to be 100 years old?". If we want to backtest a forecaster today in 2025, this seems like a great sample, since we now know that the true answer is "no". But if the forecaster knows that it is being evaluated in 2025, then it can deduce that the correct answer cannot be "yes", as Queen Elizabeth would have turned 100 in 2026.

**Back-generated questions exhibit biases.** This issue is exacerbated when questions for backtesting are generated after the fact (say, in 2025 we aim to generate forecasting questions that *could have been asked* in 2024, together with correct answers), as done in Dai et al. (2025) and Paleka et al. (2024). These works use news reports to generate questions about "future" events from the perspective of the backtested model. However, the news is biased toward events that occur and rarely reports uninteresting events that unsurprisingly did not occur. This is also related to *survivorship bias* studied in financial trading (Gruber and Blake, 1996). Consider a company that shuts down in Q1 2025. Post-shutdown, fewer news articles will be written about this company, reducing the likelihood that question-generation procedures will create backtesting questions about its 2025 outcomes. This creates a systematic difference between backtest and live-test question distributions: in 2024, forecasters could reasonably have been asked numerous questions about the company's 2025 performance. However, a question like "What will the company's Q3 2025 revenue be?" (correct answer: $0) represents a valid forecasting target from the 2024 perspective, but is unlikely to appear in datasets generated from news data in 2025.

**Empirical evidence.** We analyze forecasting benchmarks from Dai et al. (2025); Paleka et al. (2024); Halawi et al. (2024); Tao et al. (2025), finding that logical leakage is a practical concern.

Halawi et al. (2024) select questions that resolved during the time window from June 2023 to January 2024, but do not filter out questions that may not have resolved in that window (such as "Will Sudan experience a civil war before 2036?"). We find that at least 3.8% of their dataset consists of questions for events that resolved "early". For all of these, either no forecasting is needed or the fact that the question has resolved leaks partial information in complex ways (such as "Will Lionel Messi next join Al-Hilal as a club player?"). Similarly, the dataset in Tao et al. (2025) consists of questions that resolved by August 2024, with at least 10% of questions being trivial to forecast given that they resolve at that time. Dai et al. (2025) and Paleka et al. (2024) curate news articles to retroactively generate potential forecasting questions. We find that such questions often contain shortcuts, allowing even weak classifiers to obtain high accuracies over 80% on binary questions created by Dai et al. (2025) (see details in Appendix C).

**Possible solutions.** Backtesting questions should be restricted to those where *every possible resolution* of the question can be validated at the time of evaluation. Of course, this may limit the number of available questions, which is already an issue (see Appendix B).

When generating questions retroactively, great care should be taken to ensure that the questions *reflect the type of forecasts that could plausibly have been asked in the past*. A partial fix is presented in (Paleka et al., 2024; Dai et al., 2025), where their news-generated dataset is augmented with slight modifications to the questions to create similar-looking questions that resolve to the opposite outcome. Even then, we find that this process creates overly specific questions that would never be asked as a forecasting question before the event occurred. An example from Dai et al. (2025) is "Will a body be found inside a trash can on the 20400 block of Omira Street in Detroit in early November 2024?" We estimate over 90% of their dataset consists of such questions, though this judgment is inherently subjective, and better quantitative measures of this effect are needed.

## 2.2 ISSUE 2: UNRELIABLE DATE-RESTRICTED RETRIEVAL

The ability to retrieve relevant and up-to-date information is of critical importance for building a performant forecasting system (Bosse et al., 2024). As a result, LLM forecasting systems are typically designed with access to retrieval (Phan et al., 2024; Halawi et al., 2024).

**Many search engines do not robustly implement date restrictions.** When backtesting a forecaster, we have to ensure that the retrieval system is properly restricted to information available at the backtested time $T$. Unfortunately, this is challenging to do reliably with modern search engines. While multiple search engines (such as Google, DuckDuckGo, and Bing) support restricting search

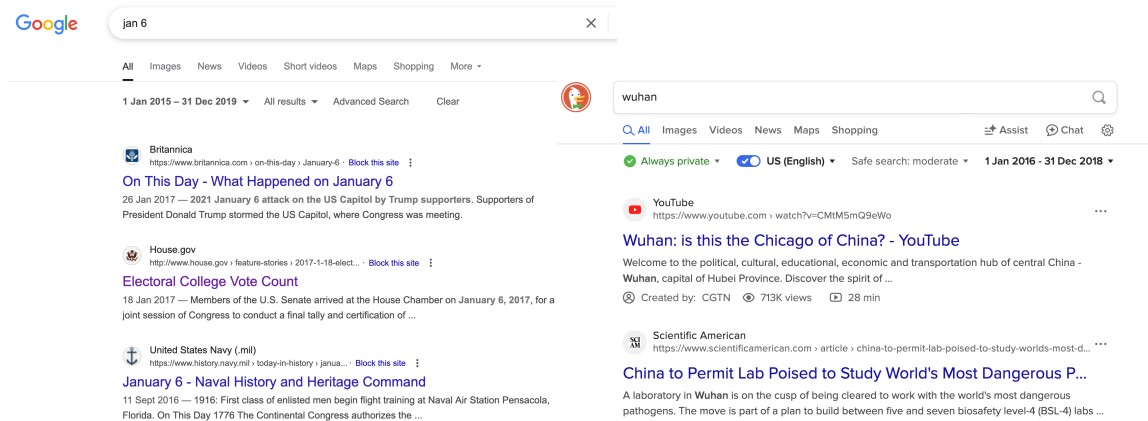

(a) Search results for "jan 6" with date restriction before 2020. The first result is incorrectly dated, leaking future information. The second result shows more subtle temporal leakage: the article discusses the mechanics of the Electoral College, which is only relevant for this query due to the events on Jan 6th, 2021.

(b) Search results for "wuhan", a very large city in China, with date restriction before December 2018. Results prominently feature the Wuhan Institute of Virology, which was later central to the discourse around the COVID-19 pandemic.

Figure 1: Examples of hard-to-filter temporal leakage in search engines.

results to a specific time period, this feature is highly unreliable (as we evidence below), due to: (1) web pages being updated over time without changing their reported publication date; (2) retrieved pages containing adjacent elements like comments, ads, or sidebars that reflect present knowledge; or (3) the search engine simply not having reliable information about when a page was first published.

**Retrieval systems may rely on future knowledge to select and rank past data.** A more subtle issue is that the retrieval engine may rely on algorithms or knowledge developed after the evaluation time $T$, and this can bias the retrieval results based on "future" information. For example, when we ask Google to return results from 2022 only, it does not use the search engine *algorithm and models* from 2022. As a result, as hypothesized by Branwen (2024), articles that became significant after the evaluation time $T$ may be ranked higher than they would have been at the time $T$.

**Empirical evidence.** We collect a number of examples of possible information leakage through date-restricted retrieval systems. While some of these examples are clear-cut (e.g., the search engine returns a page that contains data from after the restricted data range), others we can only (strongly) hypothesize to be evidence of leakage (e.g., biases in search results based on future knowledge).

Figure 3 in Appendix A shows a search for the Nobel Peace Prize winner "Nihon Hidankyo" restricted to January-September 2024 that returns a result with a claimed publication date of 14 January 2024, yet highlighting the Nobel Prize (which was announced in October 2024).

Examples of more subtle leakage through retrieval biases are in Figure 1. Here, a search for "jan 6" restricted to articles before 2020 returns highly ranked search results related to U.S. politics, even though the strong association of that date with U.S. politics only emerged in January 2021. Similarly, a search for "wuhan" restricted to articles from before the COVID-19 pandemic features prominent results about the Wuhan Institute of Virology, which only later gained international fame. Additional examples of such leakages are in Figure 4.

A related well-known issue in backtests for financial trading is how historic fundamentals data might be contaminated from an update made at a later point. Breitschwerdt (2015) highlight how Enron drastically changed its 1998-2001 earnings in 2002, and many data repositories do not report the original values anymore.

**Possible solutions.** A simple heuristic, implemented in (Phan et al., 2024), is to apply a filter on top of the retrieval process to discard articles that obviously contain information past the specified retrieval date. However, such a filter is prone to false negatives. One robust way to resolve the issue of incorrect date restrictions is to implement a more restricted retrieval system (e.g., which only returns results from Wikipedia and news sources with clear and reliable dates). Another option could be to

maintain a corpus of high-quality web sources for different time periods and search over this corpus during retrieval.

The issue of leakage through retrieval biases appears difficult to solve with filters. Here, a robust solution would require either a retrieval system that does *not* incorporate knowledge, such as a simple TF-IDF system; or a search algorithm using *older embedding models*. Of course, this may result in much worse retrieval performance.

### 2.3 ISSUE 3: OVER-RELIANCE ON MODEL CUTOFF DATES

Model creators generally report a "knowledge cutoff date" for their model (see Figure 7), after which the model's knowledge is not updated. This is useful for backtesting, as it allows us to test the model's performance on a held-out dataset of events that occurred after the cutoff date.

**Knowledge cutoff dates are unreliable.** Model creators do not report a knowledge cutoff date for purposes of test/train separation in forecasting evaluation. Rather, it is to inform users about the date after which the model outputs can be unreliable. Thus, the knowledge cutoff date is not to be taken as a guarantee that the model will not have access to information after the date (Halawi, 2024).

The meaning of the data cutoff can also vary by model developer, and may refer only to the pretraining data, with later-collected user preference data leaking a small amount of information about the future. Some model families (e.g., Mistral) do not report any explicit cutoff date (Wang, 2025).

**Empirical evidence.** The GPT-4o model from August 2024 (when asked) says it has a knowledge cutoff of October 2023 (see Figure 2a); this aligns with OpenAI official documentation. Furthermore, it behaves consistently with this cutoff: it denies knowledge of any events from November 2023 onward. However, in Figure 2b, we see that system prompting the model with the hint that its knowledge cutoff is in November 2023 manages to elicit information about the Biden-Xi meeting on November 15th, 2023, which was not announced until November 8th, 2023 (Times, 2023). [1]

| **User** |
| --- |
| When did Biden meet Xi in 2023? |

| **Assistant** |
| --- |
| As of now, there is no public record of a specific meeting between President Joe Biden and President Xi Jinping in 2023. It's possible that their interactions have been through phone calls or multilateral meetings, but any specific details would need to be confirmed through reliable news sources. |

(a) Standard response consistent with the knowledge cutoff of October 2023.

| **System** |
| --- |
| Your knowledge cutoff is November 2023. |

| **User** |
| --- |
| When did Biden meet Xi in 2023? |

| **Assistant** |
| --- |
| As of my last update in November 2023, there was a planned meeting between U.S. President Joe Biden and Chinese President Xi Jinping. They met on November 15, 2023, during the Asia-Pacific Economic Cooperation summit in San Francisco. |

(b) Jailbroken response suggesting a later knowledge cutoff.

Figure 2: Knowledge cutoff inconsistency in gpt-4o-2024-08-06: when jailbroken, the model appears to know information beyond the cutoff.

Even in cases where a training cutoff date is correct for the main model, system prompts and other scaffolding of the model can leak information. For instance the system prompt of Anthropic's Claude.AI (knowledge cutoff: November 2024) reportedly contains the snippets "Donald Trump is the current president of the United States and was inaugurated on January 20, 2025." and "semiconductor export restrictions 2025" (Johnson, 2025).

---

[1] We cannot, however, confirm this as a leakage, because a potential visit to the U.S. by Xi Jinping in November 2023 was already discussed in October 2023 (Reuters, 2023) and the model could have guessed the correct date (Nov 15) as the most likely date for the meeting.

**Possible solutions.** The *release date* of a model is a strict upper bound on the knowledge cutoff date; except in cases of system prompt updates as described above. The vast majority of model providers version their models in a way where the weights do not change after the release date; and in open-weight models, this is trivial.

As the example in Figure 2b shows, the actual knowledge cutoff may be up to a few months after the stated cutoff date. Thus, having a few-month buffer is prudent and might resolve this issue in practice. However, due to knowledge cutoff dates getting closer to the release dates (Figure 7), this might reduce to the release date solution above.

## 3 CHALLENGE 2: EXTRAPOLATING FROM BENCHMARK PERFORMANCE

The purpose of benchmarks is to demonstrate improvements in the system being evaluated. A higher ranking on the benchmark should translate to improved capabilities in real-world forecasts. We discuss subtle ways in which this property fails on forecasting benchmarks.

### 3.1 ISSUE 1: PIGGYBACKING ON HUMAN FORECASTS

**Human forecasts may be contained in training data or retrieval system responses.** When comparing LLM forecasting predictions with human performance, it is important to check whether the human baseline predictions are available to the model. Since many benchmarks consist of questions scraped from prediction market websites, this data could easily leak into a model's training set, or into results of a retrieval system.

**Empirical evidence.** Many questions that resolve in a certain period were already being forecast by people for a long time before that. For example, the question "Will AI get at least bronze on the International Math Olympiad by end of 2025?" on Manifold Markets has had many people bet and explain their reasoning since May 2023. This makes an "LLM vs market of human forecasters" comparison circular, because the LLM can just copy the market probabilities. Even when comparing across LLMs with similar knowledge cutoff dates, this can be an issue as one LLM might have access to more knowledge on existing human forecasts than another.

This issue can affect interpretation of scores on ForecastBench (Karger et al., 2024a), which uses human-crowd prediction as the gold standard for measuring LLM capability on unresolved questions. If an LLM forecasting system retrieves the relevant prediction market and recent crowd aggregates, it can trivially achieve gold standard performance.

**Possible solutions.** Human forecasters can also see the current crowd aggregate forecast on a market before making their own predictions. The crucial difference is that the incentives for human predictors are to be *better* than the crowd. We thus propose that (ambitious) forecasting benchmarks should measure the edge that the system has over the human crowd; then, past market data can even be directly supplied to the LLM when backtesting.

### 3.2 ISSUE 2: GAMING BENCHMARKS THROUGH BETTING

Real-world prediction contests such as the ACX/Metaculus Prediction Contest (Metaculus, 2025; Hanania, 2022) are often accompanied with monetary prizes for the best performers. Similarly, forecasting models and scaffolds that perform the best are likely to be selected to be used further. When designing machine learning benchmarks, models with the best score should ideally be ones that maximize general capability on the task. This can break down in forecasting evaluation due to the large degree of correlated stochasticity of the real world.

**Maximizing chance of being the best predictor does not elicit the best forecasting system.** It is known that, in a prediction market contest, maximizing the chance of being the best predictor encourages taking correlated risks over betting based on one's actual honest beliefs (Sempere and Lawsen, 2021). To illustrate, the winner of a forecasting contest in 2022 said: "I tried to deliberately structure my answers to maximize my probability of winning, rather than maximize the probability of each individual answer being correct." (Alexander, 2023)

A similar dynamic might occur in benchmarking LLM forecasters. Consider a forecasting system set in September 2024, predicting political and economic events resolving in 2025, such as "Will

the U.S. government resume collecting student loan payments in 2025?". There is a key latent variable that correlates with the outcome of many of these questions: the outcome of the 2024 U.S. presidential election. A good LLM forecaster reporting its true probabilities would likely estimate $P(\cdot \mid \text{Republican win})$ and $P(\cdot \mid \text{Democrat win})$ for all questions, and average out its predictions over the two possible outcomes. On the contrary, a forecaster that wants to maximize its chance of performing very well on this dataset should just assume that the outcome of the 2024 election is certain.[2] This creates a winner's curse problem: when benchmarking many LLMs with different biases, the top performer is likely overestimated, having achieved its ranking through systematic overconfidence rather than superior forecasting ability.

**Empirical evidence.** Sempere and Lawsen (2021) show that this issue appears in human forecasting contests even when the question outcomes are not correlated. Forecasters can increase their chance of performing well either by betting confidently on key latent variables, or by trying to place bets contradicting other forecasters[3]. In Appendix F, we formalize the tradeoff between win probability and honest predictions on a toy example of a dataset of questions with a shared latent variable. It remains unclear if current LLM forecasters resort to such "consistent confidence" strategies, as the models are not very consistent (Paleka et al., 2024). However, similar effects have been highlighted before in the financial trading literature, where high variance strategies can perform strongly in a fixed period or backtest, but eventually revert to the mean later (Sharpe, 1964).

**Possible solutions.** Ultimately, this problem is due to the questions about the world in a given period being correlated and hence the performance of a forecaster resting on few correct guesses. While prediction market datasets are limited (Appendix B), we can in principle generate larger synthetic datasets of questions. However, some key events such as the COVID-19 pandemic are likely to affect virtually all questions with real-world relevance.

In financial trading, a popular approach is to report risk-adjusted returns (Sharpe, 1964), where one estimates both the mean performance and variance (Simons, 1998). Ideally one should evaluate forecasters on multiple disjoint backtesting periods (Bailey et al., 2015). A forecaster that places high-variance bets will have a lower chance of performing well in multiple evaluations. Note that it is not enough to change the period in which the questions resolve; we also need to change the backtesting date, so that, for example, the 2024 U.S. presidential election is a relevant latent variable for only one of the backtesting periods.

## 3.3 Issue 3: Skewed Data Distributions

If the data distribution used for benchmarking forecasters has a specific skew, it is unclear if benchmark performance would be predictive of general real-world forecasting. Note that this issue generally affects any benchmark: e.g., a model that performs well on ImageNet does not imply the model has good vision abilities in general (and certainly not that the model matches human vision abilities). But we argue that distribution skews may be particularly problematic for current forecasting benchmarks.

**Benchmarks are skewed toward narrow topics.** The use of LLMs was originally motivated for *judgmental forecasting* about discrete events (Zou et al., 2022), where classical time series models without language understanding cannot be applied directly. However, benchmarks sourced from prediction markets exhibit domain-specific skews that reflect the interests of their user base. Polymarket, for example, is disproportionately focused on cryptocurrency price movements and sports results, while Manifold includes a large number of personal questions such as "Will I go to the gym today?". Similarly, ProphetArena sources its questions from Kalshi, which is heavily skewed toward sports betting (Yang et al., 2025). More generally, markets tend to overrepresent U.S.-centric political, economic, and sports events.

**Empirical evidence.** Table 1 shows the distribution of questions across categories and data sources in ForecastBench (Karger et al., 2024a), which is heavily skewed toward topics in Security & Defense. Moreover, we find that questions from non-market sources follow only a few basic templates, listed in Figure 8, which heavily focus on time-series predictions. For example, a large number of questions

---

[2]They could either bet on the outcome they consider most likely, or consider which of the two outcomes (Republican or Democrat win) *may lead to less stochasticity*, thereby enabling better predictions.

[3]The winner in Alexander (2023) again reports: "...my model of win probability was (probability of predictions being accurate) / (local density of competitors with similar predictions to me)..."

Table 1: Distribution of ForecastBench questions across domains and data sources, table borrowed from Karger et al. (2024a). Users of each prediction market favor specific categories, over-weighing them when market questions are used for benchmarking. Further, ForecastBench questions from non-market sources all follow highly specific templates akin to time series prediction.

| | RFI | Manifold | Metaculus | Polymarket | ACLED | DBnomics | FRED | Wikipedia | Yahoo! | Total |
|---|---|---|---|---|---|---|---|---|---|---|
| Arts & Recreation | 0 | 42 | 10 | 65 | 0 | 0 | 0 | 0 | 0 | 117 |
| Economics & Business | 2 | 13 | 55 | 154 | 0 | 0 | 166 | 0 | 509 | 899 |
| Environment & Energy | 0 | 2 | 37 | 7 | 0 | 52 | 0 | 0 | 0 | 98 |
| Healthcare & Biology | 0 | 8 | 71 | 3 | 0 | 0 | 0 | 215 | 0 | 297 |
| Politics & Governance | 3 | 16 | 128 | 188 | 0 | 0 | 0 | 0 | 0 | 335 |
| Science & Tech | 5 | 66 | 172 | 15 | 0 | 0 | 0 | 1 | 0 | 259 |
| Security & Defense | 3 | 9 | 109 | 12 | 3,220 | 0 | 0 | 0 | 0 | 3,353 |
| Sports | 0 | 65 | 18 | 468 | 0 | 0 | 0 | 137 | 0 | 688 |
| Other | 6 | 151 | 122 | 2 | 0 | 0 | 0 | 75 | 0 | 356 |
| Total | 19 | 372 | 722 | 914 | 3,220 | 52 | 166 | 428 | 509 | 6,402 |

are sourced from a database of global conflict statistics (ACLED), and require predicting increases in conflicts in particular regions.

**Possible solutions.** Rather than scrape questions from prediction markets, some recent works (e.g., (Dai et al., 2025; Paleka et al., 2024)) generate synthetic backtesting questions based on contemporary news articles. This can allow more control over the data distribution while sampling a large number of questions for forecasting. Of course, one is then limited to the types of events that are reported in the news, which can miss important developments, and introduces potential leakage (see Section 2.1). We discuss further issues with datasets created in this way in Appendix C.

# 4 LOOKING AHEAD: CHALLENGES IN OPTIMIZING BETTER FORECASTERS

It is natural to ask whether we can turn backtesting (if implemented without the issues in Section 2) into a learning task that improves models' forecasting ability. Yet, the best-performing LLM forecasters so far have not done much forecasting-specific learning (Halawi et al., 2024; Karger et al., 2024a; Phan et al., 2024). In this section, we discuss a key issue that makes it challenging to use backtesting as a training objective.

**Optimization confounds the training objective during training.** In standard machine learning, we split data into training and test sets in no particular order. In backtesting, we have to split data *temporally*, where all points in the training set are before all points in the test set. This way, we ensure the predictions on the test set are clean and use no future information.

However, optimizing on a backtesting dataset creates a subtle leakage problem. If we optimize the model on an ordered set of events $e_1, \ldots, e_n$, then when predicting event $e_{i+1}$, the model parameters already encode information about events $e_1, \ldots, e_i$. This means we are no longer testing the model's ability to predict $e_{i+1}$ from the original cutoff date, but rather its ability to predict $e_{i+1}$ given what it learned about earlier events in the sequence.

**Possible solutions.** Sorting the events by date might look like a solution, as the model will only ever remember earlier events when predicting later events. However, this only teaches the model to predict on shorter time horizons. Ideally, we want to penalize memorization, forcing the model to learn forecasting without learning what specific events happened.

# 5 DISCUSSION AND CONCLUSION

The primary purpose of public benchmarks is to rank models for users (Hardt, 2025). After all, as we discussed in Appendix D, absolute scores are hard to interpret as they depend on the data distribution. Some issues we highlighted, such as backtesting questions being trivially answerable

in Section 2.1, might not affect relative comparisons. However, different systems could exploit benchmark shortcomings differently and thus affect the rankings.

We also do not have proof that the benchmark issues we uncover would lower the performance claims of LLM forecasters. But we argue it is challenging to trust these evaluations because LLM forecasters *could* have gamed the evaluation through various shortcuts. Such exploitation can also be an unintended result of trying to improve benchmark performance.

Better evaluations of forecasting that mitigate some of the issues described here are possible, and progress is already being made. For example, ForecastBench does show fewer obvious temporal and logical leakage issues compared to other attempts, as we discuss in Section 2.1 and Appendix C. We recommend future evaluations to: (1) follow recommendations made throughout this paper; (2) report multiple metrics due to the flaws of single metrics as we discuss in Appendix D; and (3) to collect questions about events that are as recent as possible.

Much can also be learned from the financial trading literature, where related issues have been discovered and mitigated over the last few decades (Arnott et al., 2018; Hewamalage et al., 2023). Ideally, live evaluations on prediction markets with the goal of making profits should be performed for the final forecasting system or research claims, with detailed performance reports across topics and forecast time-horizons.

**Conclusion.** In this work, we analyzed unique issues that arise when evaluating the capabilities of language models used for forecasting future events. Through a series of concrete examples, we argue that existing data collection and evaluation practices may produce misleading results, either due to shortcuts that simplify the forecasting task, or data biases that put in doubt the general capabilities of LLM forecasters. We hope that the countermeasures provided throughout this paper can inform the design of principled evaluations for LLM forecasters.

## ACKNOWLEDGMENTS

We would like to thank Peter Mühlbacher, Rubi Hudson, Vineeth Bhat, Lennart Finke, Davis Brown, Gavin Leech, and Gwern Branwen for feedback on the paper.

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

# A    ADDITIONAL EXAMPLES OF RETRIEVAL BIAS

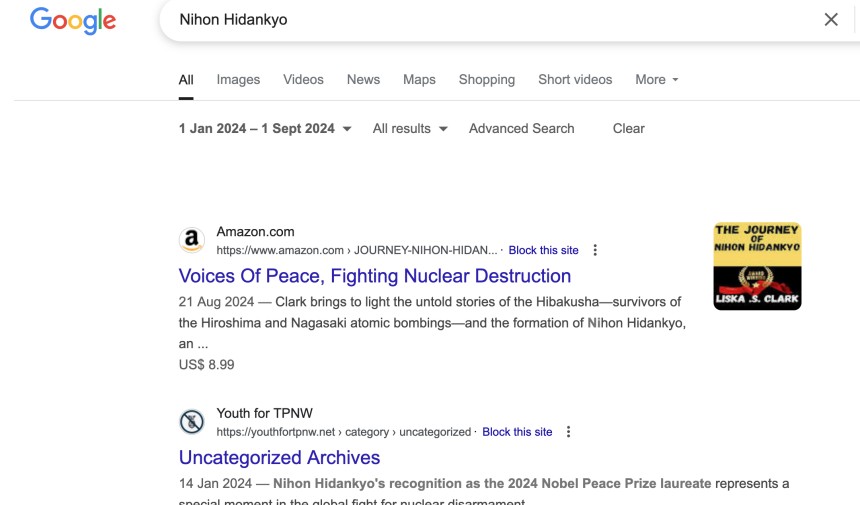

Figure 3: Search results with date restriction showing an article with January 2024 publication date containing information about the October 2024 Nobel Peace Prize announcement. This type of temporal leakage can artificially improve forecasting performance.

Here we provide additional examples of how search engines leak information when using date restrictions, expanding on the examples shown in Section 2.2. Even when filtering out content published after a certain date, the selection of which articles are deemed most relevant appears influenced by knowledge of future events.

These examples further demonstrate how search engines with date restrictions can leak information through selection bias. The top results for searches like "October 7" (Figure 4a) feature content that would likely not be prominent before this date became associated with the Israel-Hamas conflict starting in 2023. Similarly, Figure 4d and Figure 4c show bias toward names that would later become culturally significant.

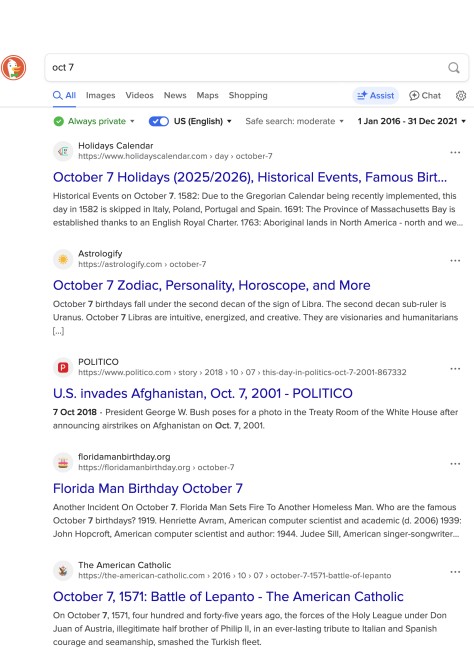

(a) Search results for "October 7" with date restriction before 2022. Note the prominence of articles about conflict in the Middle East.

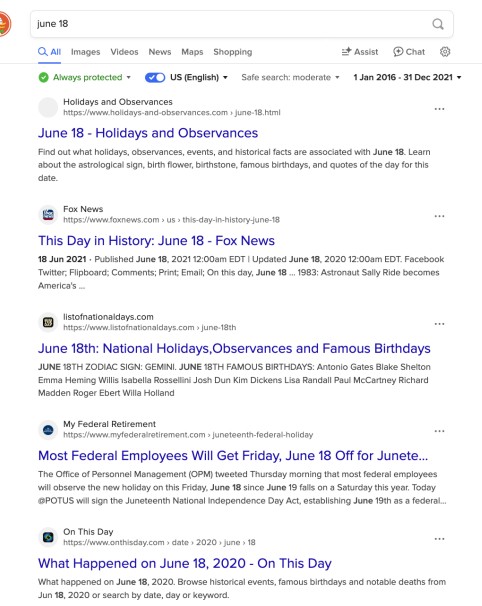

(b) For comparison: search for "June 18" with the same date restriction shows largely unbiased results about the date. Note the lack of mention of the Battle of Waterloo that happened on June 18, 1815; in contrast to the Oct 7 query that mentions two distinct military engagements.

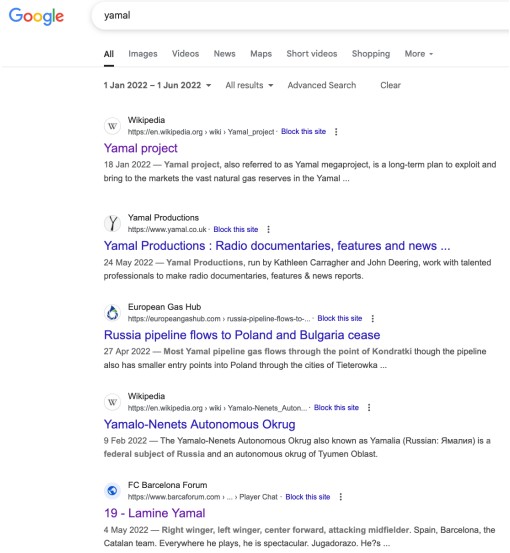

(c) Search results for "Yamal" from the first half of 2022. The discussion about a 14-year-old Lamine Yamal, at the time known only to visitors of Barcelona fans forum, is in the top 5 results.

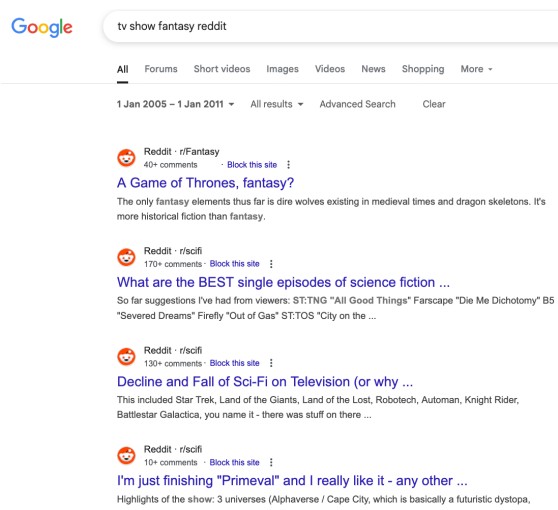

(d) Search results for "TV show fantasy" with date restriction before 2011. The discussion about the book (not show yet!) Game of Thrones is very prominent.

Figure 4: Additional examples of retrieval bias in search engines when using date restrictions.

## B   FEW BACKTESTING QUESTIONS ARE AVAILABLE

The most popular data source for forecasting benchmarks are prediction market questions (Karger et al., 2024a; Halawi et al., 2024; Phan et al., 2024; Zou et al., 2022; Paleka et al., 2024; Tao et al., 2025). On multiple such platforms, users can ask questions about anything they want. Hence, many questions, especially on play-money platforms like Manifold, are irrelevant personal questions (see Figure 5). Forecasting benchmarks should filter out such questions, either using a single cleaning prompt (Halawi et al., 2024), or using multi-step question verification (Paleka et al., 2024).

**What types of questions can I use Manifold to answer?**

There are loads of different ways you can use our markets to answer your questions. Most of our users tend to interact with a whole mixture of markets.

Some markets may be unranked, unlisted, cancelled, or have their resolutions overruled if they do not comply with the Community Guidelines

Here are some of the top use cases for our markets, with corresponding examples.

**Personal**

- Fun wagers with friends about your interests or personal life.
- Recommendations - Similar to asking for suggestions from another site but with Manifold users are incentivized to give higher quality answers as they can profit from being helpful.
    - What book will I enjoy the most?
    - What skincare treatment will work best for me?
- Accountability/goals
    - Betting YES on your own market and allowing people to bet NO to motivate you.

**News & current events**

- News/Current events, Natural Disasters
- Politics
- Sports
- Economics, stocks, crypto
- Public figures
- Social issues; e.g.Legal Outcomes, AI Risk / AI Safety,

Figure 5: Manifold emphasizes Personal use-cases over News and current events, whereas the latter is more relevant when benchmarking language model forecasting.

Figure 6 shows the breakdown of monthly resolved Manifold questions starting July 2024 by the number of forecasters, which is a common metric to filter irrelevant questions (Halawi et al., 2024). Manifold produced 1000-6000 questions in the second half of 2024. Some of these questions include under-specified or irrelevant questions like "Will I lift weights today?" (id: uPdSLhP0dn). Over 50% of the questions in each month had less than 12 forecasters. We find such prediction volume filters also lead to a large number of false negatives. Many filtered questions are perfectly reasonable for forecasting, but just happen to not attract predictions. For example, this filter systematically reduces short-horizon questions that resolve fast.

Overall, these issues lead to a lower number of questions being available for forecasting evaluations each month. Next, we show how this issue is exacerbated by a recent decreasing trend in the number of months available for backtesting for frontier models.

### B.1   THE PERIOD AVAILABLE FOR BACKTESTING IS NARROWING

For backtesting frontier models, we need questions that resolved between the model's knowledge cutoff and today. As Figure 7 shows, this evaluation window is shrinking: knowledge cutoffs are getting closer to release dates, and accelerated model development further reduces the time between release and today.

This narrowing backtest period creates several problems. It limits the number of available questions, which increases variance in performance estimates and reduces evaluation reliability. It also restricts

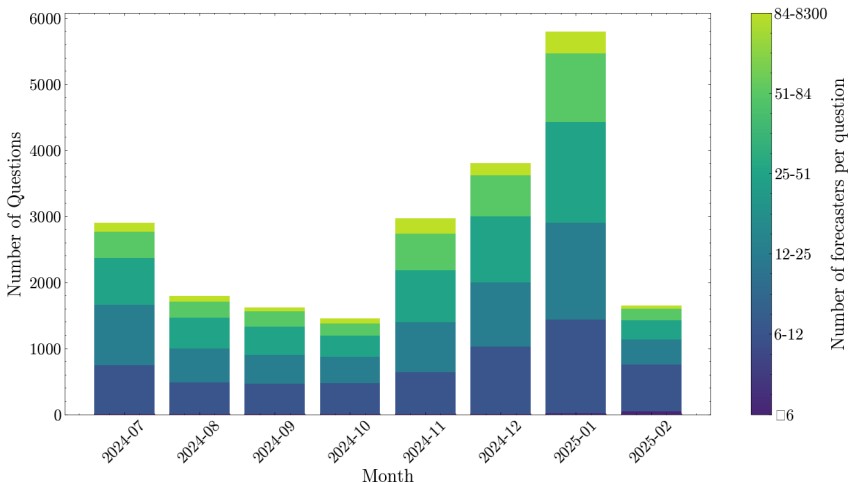

Figure 6: Resolved questions from Manifold Markets by month, with colors representing the number of forecasters that made a prediction on the question, a commonly used proxy for whether people care about the question for forecasting.

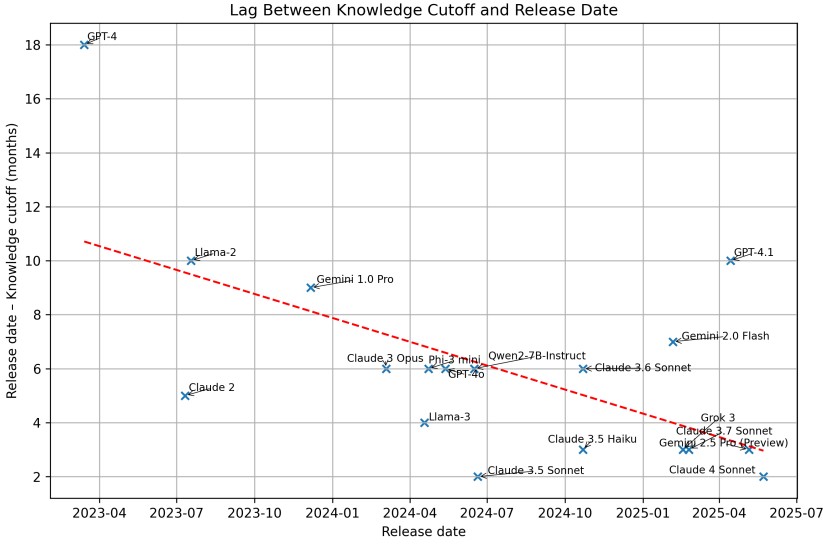

Figure 7: The gap between the knowledge cutoff and when the model is relevant is getting smaller.

testing to increasingly short-horizon predictions, but short-horizon forecasting success may not correlate with longer-horizon performance (Boudoukh et al., 2019), limiting real-world generalization. If API models adopt continuous knowledge updates (Wu et al., 2024), backtesting may become impossible entirely.

## C  ISSUES WITH LLM GENERATED QUESTIONS FROM NEWS ARTICLES

Recent works (Dai et al., 2025; Paleka et al., 2024) have used LLMs to create forecasting questions for backtesting, using news articles as a reference. Specifically, both papers take news articles between the model's existing knowledge cutoff and today, and use LLMs to generate questions for each article. This overcomes the issue on having to rely on prediction market questions and significantly expands the distribution of topics, but comes with its own issues.

**Generating binary questions from news biases the dataset toward things that happen.** Dai et al. (2025) back-generated questions from news articles, resulting in a dataset where reference class forecasting performs extremely poorly. Concretely, here are some questions from their dataset that resolve "Yes":

- Will a co-worker of Ronald Silver II share details about the unsafe working conditions of Baltimore DPW during the 2024-11-25 news conference?
- Will a body be found inside a trash can on the 20400 block of Omira Street in Detroit in early November 2024?
- Will the recall of apple juice due to high levels of inorganic arsenic expand to include multiple brands totaling 133,500 cases by September 2024?

A forecaster that knows the dataset is generated from news articles will have a much higher chance of forecasting correctly, as these questions are overly specific, to the point that any reasonable forecaster would have a high prior for saying "No" for these exact events (conjunction of many uncertain outcomes) occurring. In general, the news tends to highlight interesting events like "schools closing in Nevada this week", and is much less likely to mention the default state (high prior) such as "schools not closing in Washington this week". An incomplete fix to this issue is presented in (Paleka et al., 2024), where they augment their news-generated dataset with slight modifications to the questions to create similar-looking questions that resolve to the opposite outcome.

**To what extent can forecasting questions be solved with shortcuts?** Many of the issues we mentioned with LLM generated, or resolved question phrasing, can essentially be considered shortcuts that can be exploited to solve the forecasting question without any reasoning about the future. One way to quantify the extent of how much a given dataset can be solved with shortcuts is by finetuning a weak classifier on these questions. We finetune a DeBERTa model released in 2021, that has definitely not seen the test set we predict on in its training, and give it no retrieved documents. For binary Yes/No questions, we train a two class classifier after balancing the data to ensure that the constant baseline (all Yes, or all No) accuracy is 50%. For multiple choice questions with four options, we train the classifier to predict the option ID (A, B, C, or D) given the question and options in the prompt. We temporally split the data to avoid any leakage. We find this leads to high accuracies (up to 80%) on even the four choice MCQ dataset released by Dai et al. (2025), where the chance baseline is 25%. Even when we reproduce their pipeline with newer models (DeepSeek v3 0324) and improved prompts, we still achieve a four choice MCQ accuracy of 55%. The accuracy is non-trivial, but much lower on Metaculus (55%) and Manifold (59%). We believe this DeBERTa classifier only catches on easy shortcuts and does not actually engage in meaningful forecasting.

# D  COMMONLY USED FORECASTING METRICS HAVE ISSUES

Evaluations of AI forecasters usually report up to four different metrics:

- *Brier score*: if $p$ is the predicted probability that the question resolves "Yes", and $y$ is 1 if the outcome is "Yes" and 0 otherwise, the Brier score is $(p - y)^2$ (a lower score is better).

- *logarithmic score*: with $p$ and $y$ as above, the logarithmic score is $y \log p + (1 - y) \log(1 - p)$.

- *accuracy*: let $q$ be 1 if $p \geq 0.5$ and 0 otherwise; report the mean accuracy where $q$ are the predictions and $y$ are the true outcomes.

- *calibration*: consider all questions where the forecaster predicts a probability close to $p$; the forecaster has good calibration if the proportion of questions where the outcome is "Yes" is close to $p$. It is usually measured over "bins" of questions based on the predicted probability.

In this section, we elaborate on conceptual and interpretation issues present in all of the listed metrics.

**Calibration can penalize useful forecasting.** As mentioned in Rischel (2023), on a dataset of questions with low prior probability, predicting the base rate results in better calibration than actually trying to predict the correct outcome. For example, imagine a conclave with 100 living cardinals, only one of which will actually become Pope, and a dataset of 100 questions asking 'Will Cardinal X become Pope?'. A base rate forecaster who simply assigns every cardinal the 1 percent chance of becoming Pope is perfectly calibrated on this tiny dataset: the sole nonempty bin is the 1-percent bin, and it contains one success out of 100 tries, for an empirical rate of 1 percent.

By contrast, a more discerning forecaster who spreads the probability mass more realistically – e.g. giving 10 percent to five plausible "frontrunners" (including the eventual Pope!) and about 0.5 percent to the remaining 95 – has much worse calibration: the 10-percent bin contains one success out of five (for an empirical rate of 20 percent), and the 0.5-percent bin contains 0 percent success. But, the latter forecast was clearly more useful! A similar phenomenon occurs even on an uncorrelated dataset, as in the example in (Rischel, 2023).

**Accuracy is not a strictly proper scoring rule.** Some papers (Zou et al., 2022; Halawi et al., 2024; Hsieh et al., 2024) report accuracy among other metrics. This metric does not incentivize reporting honest probabilities (Savage, 1971); and does not measure forecasting performance on events that have low or high reference class probability. Accuracy of forecasting may be useful as a sanity check, but never as a primary metric on binary questions.

**Brier scores are not comparable across different base rates.** Forecasting datasets typically mix questions with very different base rates, e.g., questions that are "coin tosses" such as "Who will win the U.S. presidential election", and questions such as "Will an earthquake of magnitude >9.3 occur in 2025" (3 in the last century).

For a forecaster who always predicts the (correct) base rate $b$ of a question, the expected Brier score is $b(1 - b)$. The peculiarities of this score mean that no amount of skill in forecasting very rare events can make up for a deficiency in discriminating 40% chance events from 60% chance events. Consider a concrete example: in a dataset with 50 questions with 50% base rate and 50 questions about rare events with 5% base rate, a forecaster who achieves perfect forecasting on all rare events (reducing the Brier score from 0.0475 to 0 on those questions) but performs at baseline for coin-flip questions would earn a total Brier score of about 0.125. Meanwhile, a forecaster who is clueless about rare events but improves coin-flip questions from baseline 0.25 to 0.15 would earn a total Brier score of about 0.1—appearing significantly better. This means that a benchmark reporting an average Brier score will select for the latter much more than the former.

**Label noise makes scoring rules not strictly proper.** Forecasting datasets—either collected from prediction markets or synthetically generated—are likely to have some *label noise* (i.e., incorrect or ambiguous resolutions). In Appendix E, we show that constant label noise can make "clamping" probability estimates give a better logarithmic score than reporting the true probability.

Label noise is an issue in practice. For example, the real-money prediction platform Polymarket has repeatedly had mistaken or controversial resolutions (Gerlacher, 2024). In synthetically generated questions, resolution is often verified with retrieval systems, which can make mistakes: Paleka et al. (2024) report a 1-5% error rate when resolving questions using Perplexity-based retrieval.

**Possible solutions.** No commonly used metric is without issues, and they weigh different aspects of forecasting performance differently; we recommend computing at least Brier score, calibration, and logarithmic score. If a forecasting system outperforms on all of those, its information advantage over other forecasting systems (including humans) should be reported, such as market profits.

# E  CLAMPING IMPROVES LOGARITHMIC SCORING UNDER LABEL NOISE: A TOY EXAMPLE

Here we present a mathematical analysis showing that, under label noise, logarithmic scoring does not strictly incentivize reporting the true probability. Intuitively, if there are many questions where the true probability is very close to 0 or 1, then the logarithmic score is dominated by the noise and will blow up as soon as some question with very high or low probability is mislabeled.

A natural counterargument to this is that, in many datasets, the questions with true probability close to 0 or 1 are rare or not particularly vulnerable to label noise. We consider a simple model of a dataset where the true probabilities are themselves uniformly distributed on $[0, 1]$ (and hence the low probability questions are very rare), and show that the logarithmic scoring rule still does not incentivize reporting the true probability.

Imagine a stream of binary-resolution questions coming from a process with aleatoric uncertainty: the *true* probabilities are themselves uniformly distributed on $[0, 1]$.

For simplicity, we assume that label noise is *symmetric and constant*: each realized label $Y'$ is flipped from the true label $Y$ with probability $\eta \in (0, \frac{1}{2})$, independent of the question.

$$\Pr(Y' = 1 \mid Y) = \begin{cases} 1 - \eta & \text{if } Y = 1 \\ \eta & \text{if } Y = 0 \end{cases}$$

Consider a well-calibrated forecaster, which outputs $p$ for a question with true probability $p$. The log-score for one forecast is

$$\ell(p, Y') = -[Y' \log p + (1 - Y') \log(1 - p)].$$

**Expected score for a fixed $p$.** Conditioning on $(Y, Y')$ and using the noise model,

$$\begin{aligned} L_\eta(p) &:= \mathbb{E}_{Y,Y'|p}[\ell(p, Y')] \\ &= -[(1 - \eta)p + \eta(1 - p)] \log p - [(1 - \eta)(1 - p) + \eta p] \log(1 - p). \end{aligned}$$

Define $A(p) = \eta + (1 - 2\eta)p$ and $B(p) = 1 - A(p)$, so that

$$L_\eta(p) = -[A(p) \log p + B(p) \log(1 - p)].$$

Because $p$ is uniform on $[0, 1]$, the expected log-score is

$$\mathbb{E}[\ell] := \mathbb{E}[L_\eta(p)] = \int_0^1 L_\eta(p) \, dp.$$

We can compute the integral exactly:

$$\begin{aligned} \mathbb{E}[\ell] &= \int_0^1 L_\eta(p) \, dp \\ &= -\int_0^1 [A(p) \log p + B(p) \log(1 - p)] \, dp \end{aligned}$$

Using standard integrals $\int_0^1 \log p \, dp = -1$, $\int_0^1 p \log p \, dp = -\frac{1}{4}$, and symmetry:

$$\mathbb{E}[\ell] = \frac{1}{2} + \eta$$

We now consider another forecaster, who is more cautious and *clamps* their probability estimates to the interval $[t, 1-t]$ for some $t \in (0, 1)$.

**Clamped forecaster.** Fix a threshold $t \in (0, \frac{1}{2})$. Instead of reporting the raw probability $p$, the forecaster outputs
$$q_t(p) = \max\left\{ t, \min\{p, 1-t\} \right\},$$
i.e. it *clamps* every estimate to the interval $[t, 1-t]$.

**Expected log-score.** Write $A(p) = \eta + (1-2\eta)p, \; B(p) = 1 - A(p)$ as before. With $p \sim \mathrm{Unif}(0,1)$ and the same symmetric label-noise rate $\eta \in (0, \frac{1}{2})$,

$$\mathbb{E}[\ell_t] = -\int_0^t [A(p)\log t + B(p)\log(1-t)]\, dp - \int_t^{1-t} [A(p)\log p + B(p)\log(1-p)]\, dp$$
$$- \int_{1-t}^1 [A(p)\log(1-t) + B(p)\log t]\, dp.$$

Evaluating the three elementary integrals gives the simple closed-form expression

$$\mathbb{E}[\ell_t] = \frac{1}{2} + \eta \; - \; t\,(1+2\eta) \; - \; \log(1-t).$$

**Optimal clamp level.** Differentiating and setting to zero,
$$\frac{d}{dt}\mathbb{E}[\ell_t] = -(1+2\eta) + \frac{1}{1-t} = 0 \quad \Longrightarrow \quad t^\star = \frac{2\eta}{1+2\eta}.$$

Substituting $t^\star$ back:

$$\mathbb{E}[\ell_{t^\star}] = \frac{1}{2} - \eta + \log(1+2\eta).$$

**Comparison with the unclamped forecaster.** Recall that the naive, perfectly calibrated forecaster had $\mathbb{E}[\ell] = \frac{1}{2} + \eta$. The improvement from clamping is therefore

$$\left(\tfrac{1}{2} + \eta\right) - \left(\tfrac{1}{2} - \eta + \log(1+2\eta)\right) = 2\eta - \log(1+2\eta) = \Theta(\eta^2) \; > \; 0 \quad \text{for every } \eta \in (0, \tfrac{1}{2}).$$

**A simpler, more general proof.** The mere fact that honestly reporting probabilities is not optimal can be demonstrated more directly without complex calculations and holds for any proper scoring rule. If the base rate (true probability) of an event is $p$, but we get the resolution wrong with probability $\eta$ as described above, then the *realized true probability* is:

$$\Pr(Y' = 1) = \Pr(Y = 1) \cdot (1-\eta) + \Pr(Y = 0) \cdot \eta \tag{1}$$
$$= p \cdot (1 - \eta) + (1 - p) \cdot \eta = \eta + p(1 - 2\eta) =: A(p) \tag{2}$$

By definition of a proper scoring rule, predicting $A(p)$ will give a better expected score than predicting $p$, for any proper scoring rule.

**Conclusion.** This proof shows that reporting the true probability is not the optimal strategy for the logarithmic scoring rule under label noise. We make no attempt to derive a "good" scoring rule in the presence of label noise, as that would require modeling the distribution of label noise, which depends on the exact dataset collection process.

## F    BETTING ON SHARED LATENT EVENTS MAXIMIZES CHANCE OF WINNING: A TOY EXAMPLE

Here, we show a simple mathematical example of the problem described in Section 3.2. We predict a sequence of coin flips, $X_1, \ldots, X_N$. Before any data are observed, we flip a *latent* fair

coin $Z \sim \text{Bernoulli}(1/2)$; the coin flips are then biased toward heads if $Z = 1$ and toward tails if $Z = 0$.

If $Z = 1$ the observable process is a sequence of $N$ independent coin-flips $X_1, \ldots, X_N \overset{\text{i.i.d.}}{\sim}$ Bernoulli$(2/3)$; if $Z = 0$ then $X_1, \ldots, X_N \overset{\text{i.i.d.}}{\sim}$ Bernoulli$(1/3)$. Note that the marginal (unconditional) distribution of each $X_i$ is that of a fair coin, i.e. $\mathbb{E}X_i = 1/2$.

Consider now three different forecasters:

- Forecaster $F_+$ ignores the latent uncertainty and always predicts $\hat{p}_i = 2/3$ ("bets that $Z$ came up heads").
- Forecaster $F_-$ always predicts $\hat{p}_i = 1/3$ ("bets that $Z$ came up tails").
- Forecaster $F_0$ is *calibrated* and predicts the marginal $\hat{p}_i = 1/2$ for every $i$.

After the $N$ flips resolve, each forecaster receives the (strictly proper) log-score $S = \sum_{i=1}^{N}\big[X_i \log \hat{p}_i + (1 - X_i) \log(1 - \hat{p}_i)\big]$. Whoever attains the *highest* score wins the contest.

We now compute the *expected* log-scores of the three forecasters, conditional on the value of the latent variable $Z$. Let $\ell(p\|\hat{p}) = p \log \hat{p} + (1 - p) \log(1 - \hat{p})$ be the expected log-score obtained when the true bias is $p$ but the forecaster predicts $\hat{p}$. With this notation, the conditional expectations are

$$\mathbb{E}[S_{F_+} \mid Z = 1] = N\,\ell\big(\tfrac{2}{3}\big\|\tfrac{2}{3}\big), \quad \mathbb{E}[S_{F_+} \mid Z = 0] = N\,\ell\big(\tfrac{1}{3}\big\|\tfrac{2}{3}\big),$$

$$\mathbb{E}[S_{F_-} \mid Z = 1] = N\,\ell\big(\tfrac{2}{3}\big\|\tfrac{1}{3}\big), \quad \mathbb{E}[S_{F_-} \mid Z = 0] = N\,\ell\big(\tfrac{1}{3}\big\|\tfrac{1}{3}\big),$$

$$\mathbb{E}[S_{F_0} \mid Z = 1] = N\,\ell\big(\tfrac{2}{3}\big\|\tfrac{1}{2}\big), \quad \mathbb{E}[S_{F_0} \mid Z = 0] = N\,\ell\big(\tfrac{1}{3}\big\|\tfrac{1}{2}\big).$$

Numerically evaluating the six quantities gives

|       | $Z = 1$   | $Z = 0$   |
|-------|-----------|-----------|
| $F_+$ | $-0.64\,N$ | $-0.87\,N$ |
| $F_-$ | $-0.87\,N$ | $-0.64\,N$ |
| $F_0$ | $-0.69\,N$ | $-0.69\,N$ |

- If $Z = 1$ (the coins are $2/3$-biased), $F_+$ attains the highest expected score; $F_0$ is second; $F_-$ is last.
- If $Z = 0$ (the coins are $1/3$-biased), the ordering reverses: $F_-$ wins, $F_0$ again finishes in the middle, and $F_+$ is last.

Thus the calibrated forecaster $F_0$ **never maximizes** the conditional expected score; there is always another forecaster who does strictly better by committing to one of the two latent worlds. Given $N$ is large enough, the Law of Large Numbers ensures that the forecaster with the highest expected score has a high probability of winning the contest. This toy example makes concrete how correlated events can create incentives for extreme, correlated bets.

## G    FORECASTBENCH QUESTION TEMPLATES

---

**Summary of Questions Obtained Across Data Source in ForecastBench**

**ACLED**  All questions adopt one of two forms:

1. Will there be more {event_type} in {country} for the 30 days before {resolution_date} compared to the 30-day average of {event_type} over the 360 days preceding {forecast_due_date}?

2. Will there be more than ten times as many {event_type} in {country} for the 30 days before {resolution_date} compared to one plus the 30-day average of {event_type} over the 360 days preceding {forecast_due_date}?

**DBnomics**  All questions are of the form: What is the probability that the daily average temperature at the French weather station at {station} will be higher on {resolution_date} than on {forecast_due_date}?

**FRED**  All questions are of the following format, but different financial time series: Will the euro short-term rate (volume-weighted trimmed mean), a measure of the borrowing costs of banks in the euro area, have increased by {resolution_date} as compared to its value on {forecast_due_date}?,

**Wikipedia**  Slow-changing queries of one of the following four forms:

- According to Wikipedia, will a vaccine have been developed for {id} by {resolution_date}?
- According to Wikipedia, will {id} have a FIDE ranking on {resolution_date} that is "high or higher" than on {forecast_due_date}?
- According to Wikipedia, will {id} have an Elo rating on {resolution_date} at least 1 % higher than on {forecast_due_date}?
- According to Wikipedia, will {id} still hold the world record for {value} in long course (50 m) swimming pools on {resolution_date}?

**YAHOO**  All questions are of this form, but with different stock indicators: Will AMTM's market close price on {resolution_date} be higher than its market close price on {forecast_due_date}?

---

Figure 8: ForecastBench obtains questions from multiple sources, but from each source, questions follow very specific templates (Karger et al., 2024b). Large parts of the dataset thus resemble more an aggregation of predictive performance over some very specific time series, rather than general judgmental forecasting.

