# OpenReview forum: "Pitfalls in Evaluating Language Model Forecasters"
_ICLR.cc/2026/Conference — ICLR 2026 Poster_

### Official Review · Reviewer_f2Pb · 2025-10-30

**Soundness:** 3
**Presentation:** 2
**Contribution:** 2
**Rating:** 8
**Confidence:** 4

**Summary:**

The authors discuss many issues with regards to forecasting benchmarks for LLMs. They mainly argue that LLMs can potentially not be trusted for real world forecasting due to these isssues.

**Strengths:**

- Very good arguments about current issues with using LLMs for forecasting.
- Provides empirical evidence of the discussed issues.
- They propose solutions to the discussed ideas, although only as future work directions to be explored.

**Weaknesses:**

- While the authors do have evidence for their claims, it is somewhat limited. As the authors mention themselves: "We also do not have proof that the benchmark issueswe uncoverwould lower the performance claims of LLM forecasters". More damning evidence would strengthen the claims.

**Questions:**

- Page 9 weirdly ends with empty vertical space. Have the authors added a page break or vertical spacing?
- Have the authors changed the default font of the ICLR latex file? This is not something that should be changed.

---

> ### Author Response · Authors · 2025-11-19
>
> We are glad you find the arguments we bring up about issues with testing LLMs for forecasting convincing. We agree that we propose solutions; but leave the implementation of such solutions in real benchmarks for future work.
>
> Thank you for noticing the formatting issues, we have fixed them in our new revision.

---

### Official Review · Reviewer_14YB · 2025-10-30

**Soundness:** 4
**Presentation:** 4
**Contribution:** 4
**Rating:** 8
**Confidence:** 5

**Summary:**

The paper argues that evaluating LLMs as forecasters has two primary failure modes. First, backtests are hard to trust: they can leak information via “logical leakage”, unreliable date-restricted retrieval (search engines yield future-informed or misdated pages), and over-reliance on reported model cutoff dates that are only loose guidelines. Second, the paper explains why even “clean” benchmark gains may not translate to real-world forecasting skill. For example, LLMs can ingest or retrieve human crowd forecasts and then be credited with matching humans.

The paper offers practical guidance for the field moving forward, such as penalizing memorization and being cautious about turning backtests into training objectives. It also suggests that research in the field should constrain retrieval to well-dated corpora, add buffers around model cutoffs, report multiple metrics, and, where possible, include live market evaluations to substantiate claims

**Strengths:**

I find the paper very well written and the arguments are presented cleanly and convincingly. In particular, the main argument is thoughtful. It systematizes failure modes for evaluating LLM forecasters into two categories—temporal leakage in backtests and misinterpretation of benchmark gains—which is a novel perspective in the literature.

For each of the pitfalls, the paper provides concrete examples. For instance, on logical leakage, the authors show that the very fact that a question is being graded at time T rules out some outcomes (e.g., “Jan 6” queries biased toward post-2021 associations) and argue that many published results likely overstate LLM forecasting ability because models may implicitly access post-T information.

These contributions are, to my knowledge, original. Beyond critiques, the paper offers mitigation guidance.

Finally, the paper offers a comprehensive view of the field and provides a strong critique of the status quo. I believe this kind of critical examination and revisiting is healthy for the area to move forward.

**Weaknesses:**

While I find the arguments convincing, the paper could be stronger in offering more systematic and empirical studies. For example, regarding logical leakage, I wonder if the impact of that could be precisely measured (in the context of any of the previous benchmarks). Similarly, for retrieval leakage, the paper offers a few examples. However, a broader measurement could strengthen the argument.

For each pitfall, the paper suggests various potential solutions. It would be nice to implement & test some critical ones and compare them end-to-end to demonstrate improved robustness.

Finally, I wonder if the author(s) have examined https://futurex-ai.github.io/ and have any comments.

**Questions:**

Two recent benchmarks emerged after ICLR deadline: https://mirai-llm.github.io/ and https://futurex-ai.github.io/. I wonder if the author(s) could offer any comments and whether these new eval candidates suffer any of the pitfalls.

---

> ### Author Response · Authors · 2025-11-19
> **Our Response**
>
> We are grateful for your support of our work, highlighting how our perspectives are novel, supported by evidence, and critical for the area to move forward.
>
> > While I find the arguments convincing, the paper could be stronger in offering more systematic and empirical studies. For example, regarding logical leakage, I wonder if the impact of that could be precisely measured (in the context of any of the previous benchmarks).
> >
>
> We already do this! See L140-151 where we quantify logical leakage in Halawi et al. (2024) (3.8%) and Tao et al. (2024) (>10%). We obtained these numbers by manually annotating the dataset to identify logical leakage. Examples we label as having logical leakage include:
>
> - Will Sam Altman return to OpenAI as CEO before 2026?
>
> - Will a Navy ship be captured, scuttled, sunk, or critically damaged in the Black Sea before 2025?
>
> - Will Manchester United win the 2023/2024 UEFA Champions League?
>
> If the forecaster knows the date they are tested on the forecast (in this dataset, before February 2024), they can directly infer that the above questions should be answered as Yes, Yes and No respectively, as otherwise the questions could not have resolved by then for backtesting. The questions become easier to "forecast" without needing to reason about the future.
>
> > Two recent benchmarks emerged after ICLR deadline: https://mirai-llm.github.io/ and https://futurex-ai.github.io/. I wonder if the author(s) could offer any comments and whether these new eval candidates suffer any of the pitfalls.
> >
>
> MIRAI is built on GDELT, a structured dataset that categorizes (e.g. “– 042: Make a visit”, “ 070: Provide aid, not specified”) international relations events by timestamps. We think this has the following pitfalls among those we highlighted:
>
> (1) Overreliance on model cutoff dates — The MIRAI paper tests on events starting November 2023, while the models they test (GPT 4o / Turbo) have a cutoff date in October, December 2023 respectively. Given our discussion of how model cutoff dates are not guaranteed, we think testing on events from 2024 would have removed much of the uncertainty about leakage.
>
> (2) Skewed data distribution — MIRAI by design focuses on a narrow domain. Within this domain, it uses structured, almost tabular-like data. This is not the general, open-ended natural language expressible forecasting questions LLMs hold the potential to predict, which forms the focus of our work.
>
> As for FutureX, we would like to note that they explicitly acknowledge our work, and claim to try to resolve some of the pitfalls we highlighted.
>
> That said, some challenges remain even in FutureX:
>
> (1) They evaluate models on <100 live questions every week. This makes rapid model evaluation difficult, as one has to wait for multiple weeks to get reliable signal. And even then, we think the issue we highlight in section 3.2, of gaming benchmarks through strategic betting on shared latent variables, remains hard to resolve. In any given period, gambling with high confidence on a few uncertain, highly influential facts (for e.g. the winner of a major election) can completely change model rankings, and lift a lucky, highly confident forecaster to the top.
>
> (2) Skewed data distribution. Figure 4 in their paper highlights that sports and crypto events are over-sampled in their dataset.
>
> That said, we do think FutureX is a significant improvement on past forecasting evaluations, partly thanks to them addressing the other pitfalls we highlighted in our work.

---

> ### Comment · Reviewer_14YB · 2025-11-24
>
> Thank you for the comments! This is helpful, and I will recommend accept in the later phase of the discussion.

---

### Official Review · Reviewer_xNK2 · 2025-11-02

**Soundness:** 2
**Presentation:** 3
**Contribution:** 3
**Rating:** 4
**Confidence:** 2

**Summary:**

The paper analyzes and synthesizes the evaluation of LLM forecasters. It identifies two main challenges: (1) various forms of temporal leakage make it difficult to trust evaluation results; and (2) performance in evaluations is hard to extrapolate to real-world forecasting settings, thereby affecting the effectiveness of LLMs as forecasters. Drawing on concrete cases from existing work, it illustrates how these challenges impact the evaluation of LLM forecasters.

**Strengths:**

1. The paper explores an interesting topic on the evaluation of LLM forecasters. The problem is well scoped and the structure is clear. It groups the challenges into two main categories and then discusses more specific cases under each.

2. The challenges identified by the authors are valid and important, and experiments in the subsequent work [1] further confirm them.

[1]FutureX: An Advanced Live Benchmark for LLM Agents in Future Prediction

**Weaknesses:**

1. Overall, I consider this paper closer to a position paper: its core contribution lies in raising the problem, summarizing the challenges, and offering methodological recommendations, while lacking new methods, benchmarks, or rigorous, controlled experimental evaluation to substantiate the main claims.

2. The paper focuses primarily on backtesting or retrodiction, and the authors contend that “The gold standard for evaluating a forecaster involves running it on unresolved questions, waiting until the questions resolve, and then scoring the predictions” is impractical for rapid model evaluation; however, benchmarks of this type already exist [1].

[1]FutureX: An Advanced Live Benchmark for LLM Agents in Future Prediction

**Questions:**

Please refer to Weakness.

---

> ### Author Response · Authors · 2025-11-19
> **Clarifying our contribution - not just a position paper, and FutureX builds on our work**
>
> We are glad you found the challenges we highlight important. Indeed, your observation that **subsequent work [1]** builds on our observations to create an improved forecasting benchmark highlights the impact of our work!
>
> > Overall, I consider this paper closer to a position paper: its core contribution lies in raising the problem, summarizing the challenges, and offering methodological recommendations
> >
>
> We agree that our core contribution is in highlighting novel problems and challenges in forecasting evaluation methodology missed in prior work. However, we do not consider our contribution to be merely a “position paper”. Instead, we consider our work a conceptual framework that can guide future forecasting benchmark design. Such conceptual work has been accepted at ICLR recently, e.g. https://openreview.net/pdf?id=fh8EYKFKns.
>
> > The paper focuses primarily on backtesting or retrodiction, and the authors contend that “The gold standard for evaluating a forecaster involves running it on unresolved questions, waiting until the questions resolve, and then scoring the predictions” is impractical for rapid model evaluation; however, benchmarks of this type already exist [1].
> >
>
> While Section 2 is indeed focused on backtesting issues, the issues we discuss in Section 3 (on how performance on a forecasting evaluation may not translate to improved real-world forecasting capabilities) are not limited to backtesting, and can also apply to live forecasting evaluations. For instance, the issue of gaming benchmarks through strategic betting on shared latent variables in Section 3.2 is not resolved by FutureX, as it’s a problem of forecasting evaluation in general. That said, we do think FutureX does a good job of building on our work, which it explicitly acknowledges. Note also that in FutureX, it takes a week to get any feedback, and there are <100 questions per week, meaning that in practice it can take multiple weeks to get reliable, statistically significant signal. Most people in the LLM community might not consider this “rapid model evaluation” given the pace of progress. If one wants to use this feedback to develop their forecasting system, the feedback loop would be prohibitively slow.
>
> We are happy to discuss these or any new questions you may have.
>
> > [1] FutureX: An Advanced Live Benchmark for LLM Agents in Future Prediction
> >

---

> ### Author Response · Authors · 2025-11-26
>
> Hi,
>
> We replied to your review about a week ago and just wanted to check in. Did our response address your concerns?

---

### Official Review · Reviewer_Nz99 · 2025-11-03

**Soundness:** 3
**Presentation:** 4
**Contribution:** 2
**Rating:** 4
**Confidence:** 4

**Summary:**

This is a meta paper which presents a critical analysis of various evaluation approaches to LLM forecasting. Main theme is that most claims about LLMs outperforming humans on forecasting may have flawed evaluation.

**Strengths:**

S1: I think this paper is timely is quite relevant given the surge of papers in forecasting.This is an essential paper for the community. The primary focus of the paper is to bring forth the flaws and issues in evaluation and benchmarks.

S2: Paper demonstrates various issues with prior work’s evaluations such as model cut-off date, bias in retrieval, leakag. The paper reads well. And is structured for ease of understanding.

**Weaknesses:**

W1. My main concern is that this is a meta analysis paper, where main contribution is the analysis and synthesizing prior work from the lens of evaluation. It is not a new artifact in a traditional sense like algo, data, methods etc.

W2. Some claims are supported through prompts/evidence, while others, like LLMs gaming the benchmarks, etc, are extrapolated/opinionated about.

**Questions:**

NA

---

> ### Author Response · Authors · 2025-11-19
> **Clarifying our contribution - this is not a meta analysis**
>
> We are glad you found our paper’s evaluation of issues in forecasting evaluations timely and essential for the community.
>
> > My main concern is that this is a meta analysis paper, where main contribution is the analysis and synthesizing prior work from the lens of evaluation. It is not a new artifact in a traditional sense like algo, data, methods etc.
> >
>
> We do not think our work is a meta analysis as we do not synthesize existing knowledge from prior work. Our analyses, and the conceptual concerns we raise with existing forecasting evaluations are novel, in that they had not been documented in LLM forecasting literature before:
>
> (1) We quantify logical leakage across multiple forecasting benchmarks, and are the first to point out how they can arise even from seemingly innocuous widely used practices, such as collecting resolved samples from forecasting platforms (2.1).
>
> (2) We surface leakage examples through retrieval systems, which are of increasing concern in the emerging paradigm of search-augmented LLM agents (2.2).
>
> (3) Another widely used practice is relying on the model cutoff date, for which we are the first to point out it is a guideline, not a guarantee (2.3).
>
> (4) We then also go beyond backtesting issues to highlight how even live-tests for forecasting can be unreliable to extrapolate from due to fundamental issues like benchmark gaming through betting (3.2).
>
> Overall, our paper is better characterized as a novel conceptual framework that should guide the design of future forecasting benchmarks, so the community does not repeat past mistakes. Works that contribute fewer novel artefacts than this paper have been accepted at ICLR recently, e.g. https://openreview.net/pdf?id=fh8EYKFKns.
>
> > Some claims are supported through prompts/evidence, while others, like LLMs gaming the benchmarks, etc, are extrapolated/opinionated about.
> >
>
> We think our claims in Sections 2-3 are supported beyond being opinions; every subsection contains an “Empirical evidence” paragraph. Regarding benchmark gaming, we in fact make a theoretical claim about the incentive for strategic gambling on shared latent variables. We support this with a formal example in Appendix F, and also draw on evidence from existing prediction market contests and financial forecasting that this issue has affected prior attempts at forecasting evaluations in other settings.
>
> We hope these clarifications increase your support of our work. We are happy to discuss further if you have any questions.

---

> ### Author Response · Authors · 2025-11-26
> **Does our response address your concerns?**
>
> Dear reviewer,
>
> We replied to your review about a week ago and just wanted to check in. Did our response address your concerns?

---

### Author Response · Authors · 2025-12-04

We thank the reviewers for their time and feedback. As only one reviewer had time to engage with our response before the OpenReview incident, we hope this short summary of the discussion and our outlook on it helps the new Area Chair.

We are glad all reviewers recognized our paper as clearly and convincingly describing important issues in existing evaluations for LLM forecasters. `Reviewer f2Pb` notes we provide *“very good arguments and empirical evidence about current issues with using LLMs for forecasting”.* `Reviewer 14YB` notes how our *“contributions are original”*, and *“the paper offers a comprehensive view of the field and provides a strong critique of the status quo. I believe this kind of critical examination and revisiting is healthy for the area to move forward.”* `Reviewer xNK2` highlights that *“The challenges identified by the authors are valid and important”.*

`Reviewer Nz99` emphasizes that *“this paper is timely and quite relevant given the surge of papers in forecasting. This is an essential paper for the community.”* Their only concern was about the nature of our paper, as not a typical “methods” paper. We clarified that our paper provides a novel conceptual framework to guide the design of LLM forecasting evaluations. Indeed, as `Reviewer xNK2` also noted, our work has already had an impact on subsequent forecasting benchmarks like FutureX. `Reviewer 14YB` acknowledged our analysis of newer benchmarks like MIRAI and FutureX under our framework by saying they will *“recommend acceptance in the later phase of the discussion”*.

---

### Meta-Review · Area_Chair_ZPpN · 2026-01-03

**Summary:**

All reviewers agreed the paper is well-written, timely, and valuable as a critique of current LLM forecasting evaluations. The two negative reviewers characterized the work as closer to a position paper, but after discussing with the SAC, I agree position-style contributions can be appropriate given that there is some amount of technical contributions.

On the other hand, many reviewers expressed their feelings that the paper's technical contribution and empirical substantiation can be improved. The two positive reviewers asked for more systematic measurements and/or validation (e.g., demonstrating how much published performance claims or model rankings change under stronger controls), and although it would be too much additional work to create a completely new forecasting benchmark that handles the issues properly, it may be much easier to propose a filtered down benchmark that removes the problematic questions from existing benchmarks, and check if LLM rankings change or not. Even if the rankings do not change, that will not degrade the contributions of this paper. Hence a slight revision that adds more empirical validation of the claimed impacts can be helpful. I encourage the authors to add this to the camera ready if possible.

Finally, note on minor anonymization concern: language such as "our work has already had an impact on subsequent forecasting benchmarks like FutureX" and "we do think FutureX does a good job of building on our work, which it explicitly acknowledges" can potentially reveal author identity. While this did not affect the AC's assessment, authors (and reviewers, in this case) should be more careful about anonymity and double-blindness.

**Reviewer Concerns:**

I down-weighted the concern about being a position paper after discussing with SAC. The concern that the paper does not provide sufficiently systematic/controlled experiments showing that the identified pitfalls lower performance claims or change model rankings remains. If the camera ready version of this paper can add a modified version of a previous forecasting benchmark, etc., as a deeper empirical investigation, that would make this an even better paper.

**Reviewer Scores:**

The initial scores are borderline. The two fours could have been unchanged or improved.

---

### Decision · Program_Chairs · 2026-01-26

Accept (Poster)